# Biocontrol of Maize Weevil, *Sitophilus zeamais* Motschulsky (Coleoptera: Curculionidae), in Maize over a Six-Month Storage Period

**DOI:** 10.3390/microorganisms11051261

**Published:** 2023-05-11

**Authors:** Mohamed Baha Saeed, Mark D. Laing

**Affiliations:** 1Discipline of Plant Pathology, School of Agricultural, Earth and Environmental Science, University of KwaZulu-Natal, Pietermaritzburg 3200, South Africa; 2Department of Crop Protection, Faculty of Agriculture, University of Khartoum, Khartoum 11115, Sudan; 3Center of Excellence for Pesticides and Plant Health, Faculty of Agriculture, University of Khartoum, Khartoum 11115, Sudan

**Keywords:** *Beauveria bassiana*, post-harvest, grain pests, biological control, *Sitophilus zeamais*

## Abstract

Food security is contingent upon increasing crop yields but also upon reducing crop losses to post-harvest pests and diseases. Weevils are particularly important agents of post-harvest losses in grain crops. A long-term evaluation of a biocontrol agent, *Beauveria bassiana* Strain MS-8, at a single dose of 2 × 10^9^ conidia kg^−1^ of grain was formulated in kaolin as a carrier at levels of 1, 2, 3, and 4 g kg^−1^ of grain and screened against the maize weevil, *Sitophilus zeamais*. After six months, the application of *B. bassiana* Strain MS-8 at all levels of kaolin significantly reduced the maize weevil populations compared to the untreated control (UTC). The best control of maize weevil was observed in the first 4 months after application. Strain MS-8 applied in a kaolin level of 1 g kg^−1^ performed the best, resulting in the lowest number of live weevils (36 insects/500 g of maize grain), the lowest level of grain damage (14.0%), and the least weight loss (7.0%). In the UTC the number of live insects was 340 insects/500 g of maize grain, the level of grain damage was 68.0%, and weight loss was 51.0%.

## 1. Introduction

Post-harvest pests threaten global food security by reducing the quantity and quality of grains, and by enhancing the colonization of grain by mycotoxigenic fungi. Losses are estimated at 40–90% for small-scale farmers for both cereal and legume grains. Hence, the most efficient way to stabilize food security may be to reduce post-harvest crop losses, especially to grain pests.

Maize (*Zea mays* L) is one of the most important cereal crops worldwide. In Sub-Saharan Africa (SSA) maize is a staple food, a cash crop, and a major source of calories [1,2]. In developing countries, maize provides up to 30% and 60% of the protein and energy requirements for humans, respectively [3]. Maize grain storage is crucial, in order to maintain a constant food supply throughout the year. For small-scale farmers in Africa, the main purposes of storage are to ensure household food supplies and seed for planting in the subsequent season [4,5]. However, maize storage is affected by various biotic constraints, the most important being insects, which destroy approximately 20% to 50% of stored maize in most African countries [6,7]. In addition to damage of grains by feeding, insects also cause an increase in grain temperature and moisture content, which leads to increases in seed respiration, and consequently, they pre-dispose the grain to secondary attack by fungal pathogens, such as *Aspergillus flavus*, leading to the production of mycotoxins [8,9,10,11,12].

The maize weevil, *Sitophilus zeamais* Motschulsky (Coleoptera: Curculionidae), is a serious pest of maize grain in Africa. The post-harvest losses due to the maize weevil have been recognized as an important constraint to food security. In severe infestations, maize weevils can cause losses of 90.0% [13]. Chemical control, using synthetic insecticides and fumigants, has been the mainstay of control of stored-grain insects [14,15,16]. Several groups of insecticides have been used as grain protectants, including pyrethroids, organophosphates, and carbamates [17]. However, contamination of grain with chemical residues, developing resistance to the chemicals by key pests, and concerns over the environmental impact of agricultural inputs have led to a search for novel, biologically based control measures, with few or no environmental hazards [18,19,20].

The use of entomopathogenic fungi (EPF) and other microbial control agents for the control of stored product pests is a promising strategy with the potential to eliminate the adverse effects of insecticides [21,22]. *Beauveria bassiana* (Balsamo) Vuillemin is one of the most widely used EPFs because of its proven efficiency against a wide range of pests. It has been used to control stored grain pests, such as the larger grain borer, *Prostephanus truncates* Horn (Coleoptera: Bostrichidae) [22], the rice moth, *Corcyra cephalonica* Stainton (Lepidoptera: Pyralidae) [23], the red flour beetle, *Tribolium castaneum* Herbst (Coleoptera: Curculionidae) [24,25,26], and the granary weevil, *Sitophilus granarius* Linnaeus (Coleoptera: Curculionidae) [27].

Commercial formulations of *B. bassiana* are available and have been registered by the U.S. Environmental Protection Agency (EPA) and other regulatory bodies globally for a wide range of insect control applications [28]. However, no commercial products of EPFs are currently registered for the biocontrol of stored-grain insects [29]. The long-term evaluation of formulated EPFs has been conducted against various stored grain pests with various formulations [6]. However, the use of a dry formulation of the conidia of *B. bassiana* in an inert dry carrier such as kaolin against *S. zeamais* in stored maize has not been investigated in long-term trials. Kaolin is an excellent carrier of *B*. *bassiana* conidia because of its low moisture content, its neutral action on *B. bassiana*, and a slight enhancement of the activity of *B. bassiana* as a biocontrol agent, relative to other carriers, such as corn flour [30]. However, observation by scanning electron microscope (SEM) showed that kaolin abraded parts of the waxy layer of the integuments whenever the insects moved in the grains treated with kaolin. This would make it easier for the fungus to attach to and infect weevils through their exoskeleton. This observation may explain the synergy between this carrier and the fungal conidia [30], a result similar to that found by others [31]. Prior research showed that when applied as dust to maize grain at levels below 4 g per kg of grain, kaolin did not have any direct insecticidal effects on the maize weevil [30]. Therefore, the present study aimed to evaluate the efficacy of a selected strain of *B. bassiana*, MS-8, using kaolin as the carrier, against *S. zeamais* in stored maize in an extended trial that would reflect the storage period of maize by small-scale farmers in the off-season.

## 2. Material and Methods

### 2.1. Insect Rearing

The initial population of *S. zeamais* was obtained from the Department of Plant Pathology, School of Agricultural, Earth, and Environmental Science (SAEES), University of KwaZulu-Natal. For each treatment “plot”, 500 g of yellow maize grain (Smith Animal Feed, Pietermaritzburg, South Africa) were placed into a 1 L glass jar (Victoria Packaging, Pietermaritzburg, South Africa). The seed morphology was intermediate between flint and dent, sourced from Brazil, but the cultivar was unknown. Approximately 200 unsexed adult insects were released into the jar, which was then covered with an insect net (FilmFlex Plastics Natal, Durban, South Africa) to facilitate air circulation. After 10 days of ovipositioning, all adult insects were removed using a sieve. The sieved grains were introduced into a clean jar and kept for a period of 35 days for progeny emergence. Emerging juveniles of *S. zeamais* were removed daily and transferred onto fresh maize grain in a glass jar covered with insect net, and were kept under controlled conditions (28 ± 2 °C and 50 ± 5% RH) until sufficient numbers of *S. zeamais* were collected.

### 2.2. Grain Preparation

Untreated yellow maize grain was used in this study. Grain was sieved to remove any dirt, dust, or broken grain. Any prior infestation of pests in the grain was eliminated by putting the grain in an oven at 40 °C for 4 h [32]. Disinfested grain was kept in a freezer at approximately −1 °C to protect it from any new infestations. For experimental purposes, the grain was removed from the freezer and allowed to acclimatize to ambient temperature and relative humidity, and then sun-dried.

### 2.3. Production of Conidia of B. bassiana Strain MS-8

The fungus *B. bassiana* Strain MS-8 used in this study was isolated from soil samples from Ukulinga Research Farm, Pietermaritzburg, South Africa using *Galleria mellonella* L. (Lepidoptera: Pyralidae) as a live insect bait [33]. The fungus was identified morphologically according to Zimmermann [34], using a light microscope (Zeiss Axiophot) and using PCR by Inqaba Biotech, South Africa (https://inqababiotec.co.za (accessed on 18 July 2022)). The primers used were ITS1 (5′-TCCGTAGGTGAACCTGCGG-3′) and ITS 2 (5′-GCTGCGTTCTTCATCGATGC-3). The accession number with Inqaba Biotech was NR 111594.1. The fungal isolate was not deposited in a public mycological collection because it is being commercially developed.

A procedure documented by Gouli et al. [35], with modifications, was used to prepare the dry conidia. The initial production of the fungus was conducted on potato dextrose agar (PDA) for 14 days at 28 °C. Mature conidia were collected from the surface of the medium. A conidial suspension was adjusted to 1 × 10^−8^ conidia mL^−1^ using a Neubauer Improved Hemocytometer (Hirschmann^®^, Pietermaritzburg, South Africa). Rice grain (1.5 kg) was washed and soaked overnight, before dispensing it into three sterile autoclavable bags (305 × 660 mm) (500 g per bag) [(Whitehead Scientific (Pty) Ltd., Unit 9, Van Biljon Industrial Park, Winelands Close, Strickland, 7530, South Africa)]. These were autoclaved at 121 °C for 15 min, followed by a 24 h cooling period at 22–25 °C. A suspension of conidia (50 mL of 2 × 10^9^ conidia mL^−1^) was used to inoculate the sterilized rice grain using a medical syringe. After inoculation, the contents of the bags were mixed to ensure an even distribution of the fungal suspension among the rice grains. The bag was kept for 15 days at 25 °C. Each bag was closed using a stopper for access to air. The stopper (diameter of 10 cm; length of 6 cm) consisting of two layers of tissue paper which were covered loosely with aluminum foil 4 days after inoculation, the aluminum foil was removed to facilitate air circulation. The tissue paper was removed after 10 days to expedite fungal sporulation. After 15 days, the fungal biomass in the bags was spread evenly onto paper towels in a layer approximately 10 mm deep and air-dried for 4–7 days on a laminar flow bench. The dried conidia were harvested by sieving the biomass with a 100 mm diameter sieve (150 µm pore size) and were stored at 4 °C for subsequent use.

### 2.4. Grain Treatments 

*B. bassiana* strain MS-8 was used at a single dose of 2 × 10^9^ conidia kg^−1^ of grain. The conidia were formulated in kaolin [(Kaolin White from Serina Trading (www.kaolin.co.za accessed 2 September 2022), which has a chemical composition of SiO_2_ (45%) + Al_2_O_3_ (36%)], with trace amounts of Fe_2_O_3_, TiO_2_, CaO, MgO, Na_2_O, and K_2_O. The kaolin was used at levels of 1, 2, 3, and 4 g kg^−1^ of grain. The minimum level of kaolin used of 1 g kg^−1^ of grain was chosen because prior experimentation had shown that this was the minimum quantity of kaolin that could be used to ensure even coverage of all the grains when mixing the biocontrol agent and its carrier with the maize grain. A control treatment with a zero level of *B. bassiana* (i.e., 100% kaolin with no *B. bassiana* conidia) was not run because prior research had shown that kaolin at 4 g kg^−1^ or less caused zero mortality levels in the target pests [30]. A control treatment of 100% pure conidia of *B. bassiana* without kaolin was not tested because *B. bassiana* conidia clump together and will not spread evenly onto the grain without a carrier. Secondly, the dose of 2 × 10^9^ conidia kg^−1^ of grain that was used in this study weighs 0.03 g, and this minute quantity of material could not be spread evenly over 1.0 kg of grain. Levels of kaolin lower than 1 g kg^−1^ of grain were not tested because prior experimentation showed that at least 1 g of kaolin kg^−1^ of grain was required to ensure an even, complete coverage of all grains in each sample. The chosen levels of kaolin and conidia of *B. bassiana* strain MS-8 were combined in a Petri dish, which was then sealed with parafilm, and mixed by shaking by hand. A fifth treatment was an untreated control (UTC). The treatments were replicated three times.

For all treatments, 5 kg samples of yellow maize grain were used. The conidial powder formulations and maize grain were mixed using a wooden spoon to ensure an even distribution of the biocontrol agent. To provide an initial infestation of weevils, 20 randomly selected adult insects of both sexes were released into each of the fifteen 17.5 L plastic boxes (Basix Plastics, Pietermaritzburg, South Africa) containing the treated maize grain.

### 2.5. Data Collection

Every 30 days for a period of 180 days, a 500 g sub-sample was removed from each replicate per treatment (i.e., 3 replicates per treatment), and the parameters of each replicate were used for the analysis of variance. Each sub-sample was weighed and sieved through a 300 mm diameter × 2.0 mm aperture sieve (Shalom Laboratory Supplies, Durban, South Africa). The number of live insects, number of damaged grains, weight of damaged grain, number of undamaged grains and weight of undamaged grain were recorded. Grain samples and insects were returned to the respective boxes after assessments. The number of damaged grains was expressed as a percentage of the total number of grains showing signs of insect feeding. Grain weight loss percentage was assessed using the method described by Gwinner et al. [36]:Weight loss (%) = (Wu × Nd) − (Wd × Nu) × 100) ⁄ (Wu × (Nd + Nu)
where, Wu = Weight of undamaged grain, Nu = Number of undamaged grains, Wd = Weight of damaged grain, and Nd = Number of damaged grains.

After 6 months of weevil infestation, the weight of edible grain g^−1^, percentage of edible grain remaining (%), weight of highly damaged grain plus small fractions g^−1^, dust g^−1^, and insect bio-mass (g) were evaluated for each replicate per treatment (1, 2, 3, and 4 g of kaolin kg^−1^ of grain).

The experiment was arranged in a randomized complete blocks design. Data was subjected to repeated measures analysis of variance (ANOVA) using GenStat for Windows, 17th Edition [37]. Means were compared using Fisher’s least significant difference (LSD) at a 5.0% level of significance. All figures were generated using SigmaPlot 10.0.

## 3. Results

### 3.1. Numbers of Live Insects

There were highly significant differences between time (F = 72.51; *p* < 0.001; d.f. = 5) and treatments (F = 57.41; *p* < 0.001; d.f. = 4), in their effects on the number of live weevils. However, the interaction between time and treatments was not significant (F = 0.88; 176: *p* > 0.01; d.f. = 20). Application of strain MS-8 at all levels of kaolin caused significantly reduced numbers of live insects over 6 months (Figure 1 and Figure 2). All biocontrol treatments reduced the number of live insects counted in the first 4 months. However, a slight increase in insect numbers was observed by months 5 and 6 (Figure 2). After 6 months, no significant differences were observed between the four levels of kaolin (1, 2, 3, and 4 g kg^−1^ of grain). However, treatment with Strain MS-8 at a kaolin level of 1 g kg^−1^ of grain resulted in the smallest number of live insects, with 36 insects in 500 g of maize grain. The largest number of live insects was 340 insects in 500 g of maize grain in the UTC.

### 3.2. Grain Damage (%) 

Highly significant differences were observed between time (F = 50.29; *p* < 0.001; d.f. = 5) and (F = 77.79; *p* < 0.001; d.f. = 4) in their effects on the levels of grain damage. The interaction treatments between time and treatments were also significant (F = 0.88; *p* < 0.01; d.f. = 20). Treatments of maize grain with strain MS-8 at all levels of kaolin significantly reduced the level of grain damage caused by *S. zeamais*. In the first 4 months, maize treated with strain MS-8 at all levels of kaolin developed a low level of grain damage. However, a slight increase was observed after 5 and 6 months of storage. After 6 months, similar performances in reducing the levels of grain damage were observed for all levels of kaolin. However, the treatment with strain MS-8 in kaolin at 1 g kg^−1^ resulted in the lowest level of grain damage of 14.0%, compared to grain damage of 68.0% in the UTC (Figure 3).

### 3.3. Grain Weight Loss (%)

Highly significant differences were observed for time (F = 41.97; *p* ˂ 0.001; d.f. = 5) and levels (F = 60.15; *p* < 0.001; d.f. = 4) for their effects on grain weight loss caused by *S. zeamais*. The interaction between time and levels (F = 2.31; *p* < 0.007; d.f. = 20) was also significant. A lower level of weight loss was observed after treatments with strain MS-8 at all levels of kaolin than the UTC. In the first 4 months, the application of strain MS-8 at all levels of kaolin resulted in a weight loss of ˂2.0% compared to a weight loss of 21% in the UTC. After 6 months, a low weight loss of ≤10.0% was observed as a result of the MS-8 treatments compared to a weight loss of 51.0% in the UTC. A doubling of weight loss from 27.0% to 51.0% was observed in the UTC in the spring (August/September). Statistically, the four levels of kaolin (1, 2, 3, 4 g kg^−1^ of grain) were equally effective as carriers of the MS-8 conidia, with no significant differences in grain weight loss as a result of these treatments (Figure 4).

### 3.4. Evaluation of Remaining Components

After 6 months of weevil infestation, more edible grain remained after all biocontrol treatments than the UTC. The greatest quantity of edible grain, 4700 g, resulted from the treatment of kaolin at 1 g kg^−1^, compared to 2900 g in the UTC (Table 1). Significantly (*p* ˂ 0.001) more grain dust (90 g) was obtained from the untreated maize grain compared to all treated maize grain. Losses of >2000 g were estimated from the weights of dust plus small grain particles in the UTC. The most insect biomass (33 g) was measured in the UTC compared to a mean insect biomass of 8 g after MS-8 treatments, a fourfold reduction (Table 1). The percentage of edible grain remaining was 58% for the UTC, compared with 94% after treatment of strain MS-8 in kaolin at 1 g kg^−1^. Hence, for a farmer, 36% more grain would be available to eat or to sell. The alternative perspective is that the postharvest loss of maize grain to maize weevils over 6 months would have diminished from 42% to 6%, a 700% reduction in losses.

## 4. Discussion

A significant reduction of ≥10.0% in the numbers of live *S. zeamais* adults was observed after treatment of maize grain with *B. bassiana* MS-8 at all levels of kaolin in all months. This indicates that strain MS-8 remained active over 6 months and that the mortality of emerging weevils in treated grain was ≥90.0% in most months. These results are similar to those of Kim et al. [38], who reported that most *B. bassiana* ERL836 conidia remained viable (>85%) with insecticidal activity for 18 months at temperatures of 25 and 30 °C. Their study was for three times as long as this study, which shows how durable the conidia of *B. bassiana* can be. This is crucial for farmers in developing countries who typically store their maize grains for up to 12 months between harvests. Bourassa et al. [39] observed that *B. bassiana* IMI330194 caused an even higher mortality level of 100% in *Prostephanus truncatus* (Horn) (Coleoptera: Bostrichidae) larvae. Mul et al. [40] observed that the fungus caused secondary epizootics, enhancing the impact of the biocontrol agent. This contrasts with agrochemicals or plant extracts, where the active ingredients degrade and lose activity over time. In addition, fungal infections may cause a decrease in the oviposition period and fecundity in the target insects, resulting in a reduction in the rate of increase of their population [41]. In contrast, an exponential increase in the population of live weevils was observed in the untreated maize grain in this study because, under conducive conditions, maize weevils have an accelerated growth and reproduction cycle, increasing the number of generations produced during the grain storage period, which compounds the level of damage caused to maize grain.

After 6 months, strain MS-8 at a kaolin level of 1 g kg^−1^ of grain caused the lowest level of grain damage (14.0%) compared to grain damage of 68.0% in the UTC. The high levels of grain damage recorded for the UTC, together with the reproduction of the weevils, would have caused an increase in grain temperature and moisture content. This would have pre-disposed the damaged grains to secondary attack by pathogens such as *A. flavus*, which produce mycotoxins [8,9,10,42]. The application of strain MS-8 at all levels of kaolin resulted in significant reductions in the loss of edible grain (6–10%), compared to a loss of edible grain of 58.0% in the UTC.

The evaluation of other grain components indicated that treatments with strain MS-8 at all levels of kaolin resulted in higher levels of edible grain remaining, lower levels of dust production, smaller waste fractions, and lower levels of insect biomass. Padin et al. [43] applied 50 times more conidia of *B. bassiana* (1 × 10^10^ conidia kg^−1^ of grain) than used in this study but reported similar results, with *B. bassiana* applications significantly reducing loss of wheat grain infested with *S. oryzae*.

In conclusion, *B. bassiana* strain MS-8 formulated in four levels of kaolin induced a high level of mortality in *S. zeamais*, which resulted in reduced levels of grain damage and grain weight loss over a 6-month storage period. This suggests that *B. bassiana* strain MS-8, formulated as a dry product in kaolin, could be used for the management of maize weevils in maize grain in storage, following its registration and commercialisation. This study showed that the use of biocontrol agents for the control of post-harvest grain pests can provide long-term protection of stored grain crops, which would enhance food security, especially in developing countries.

## Figures and Tables

**Figure 1 microorganisms-11-01261-f001:**
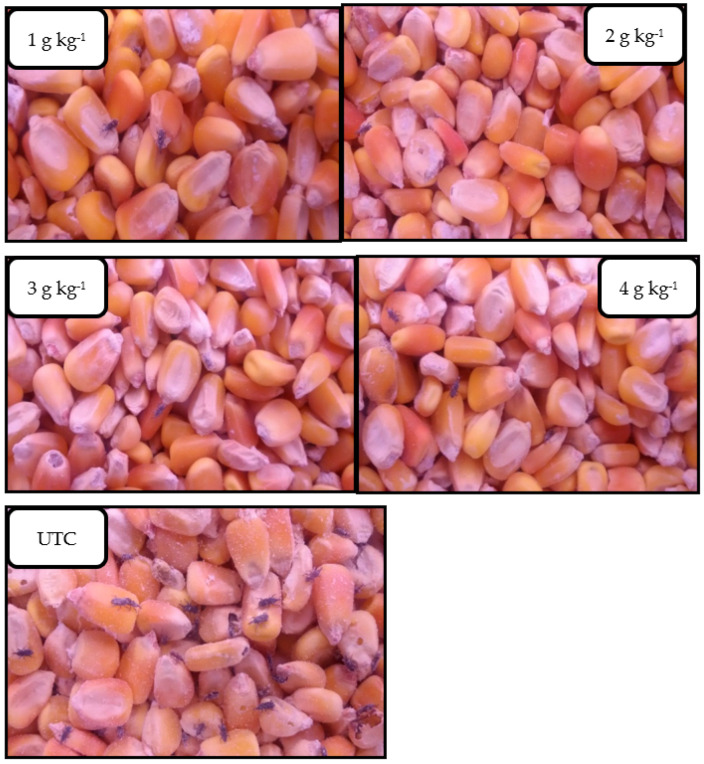
The visual condition of maize grain after infestation with *Sitophilus zeamais* for 6 months, following treatment with a fixed dose of *Beauveria bassiana* strain MS-8 (2 × 10^9^ conidia kg^−1^ of grain) formulated in kaolin at levels of 1, 2, 3, and 4 g kg^−1^ of grain, compared to an untreated control (UTC).

**Figure 2 microorganisms-11-01261-f002:**
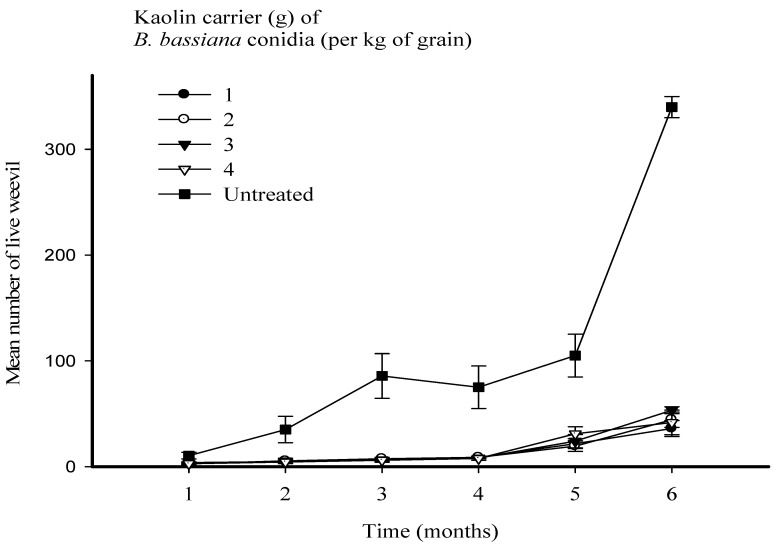
Mean number of live maize weevils (*Sitophilus zeamais*) every month for a period of six months in 500 g of maize grain, following treatment with a fixed dose of *Beauveria bassiana* Strain MS-8 (2 × 10^9^ conidia kg^−1^ of grain), formulated in kaolin at levels of 1, 2, 3 and 4 g kg^−1^ of grain, compared with an untreated control. The error bars represent the SE.

**Figure 3 microorganisms-11-01261-f003:**
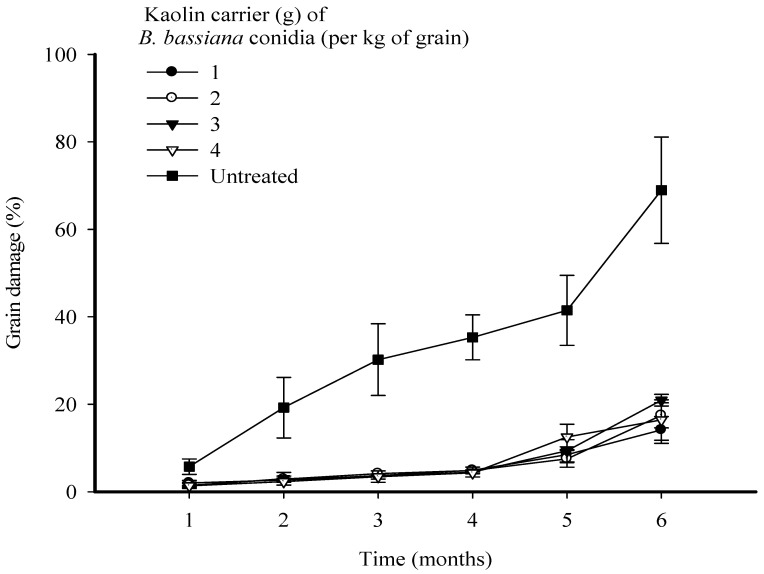
Grain damage every month over a period of 6 months of *Sitophilus zeamais* infestation of maize grains following treatment with *Beauveria bassiana* strain MS-8 at a fixed dose of 2 × 10^9^ conidia kg^−1^ formulated in kaolin at levels of 1, 2, 3, and 4 g kg^−1^ of grain. The error bars represent the SE.

**Figure 4 microorganisms-11-01261-f004:**
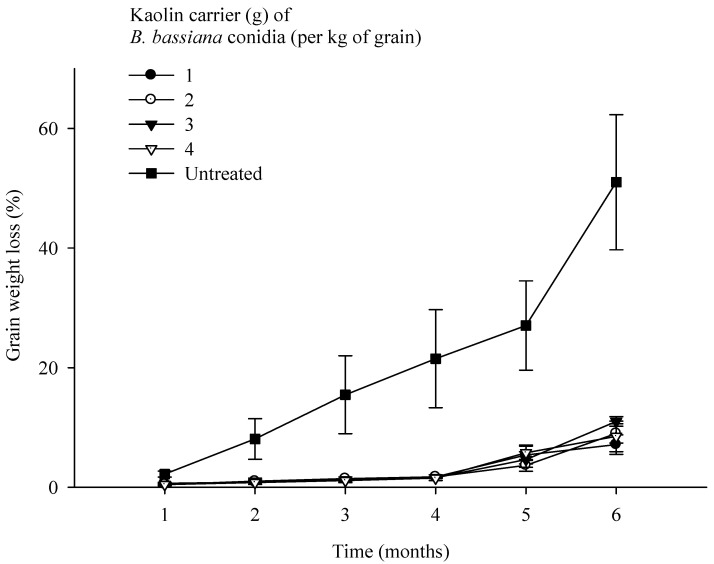
Grain weight loss every month over a period of *Sitophilus zeamais* infestation of maize grains following treatment with a fixed dose of *Beauveria bassiana* Strain MS-8 (2 × 10^9^ conidia kg^−1^ grain) formulated in kaolin at levels of 1, 2, 3 and 4 g kg^−1^ of grain. The error bars represent the SE.

**Table 1 microorganisms-11-01261-t001:** Grain fractions after six months of *Sitophilus zeamais* infestation of maize grains.

Kaolin Levels	Initial Grain	After Six Months of Storage				
		Edible Grain g^−1^	Edible Grain Remaining (%)	Highly Damaged Grain + Small Fractions g^−1^	Dust g^−1^	Insect Bio-Mass (g)
1 g	5000 g	4700 b	94 b	278 a	15 b	7 b
2 g	5000 g	4600 b	92 b	375 b	17 b	8 b
3 g	5000 g	4500 b	90 b	471 c	20 b	9 b
4 g	5000 g	4650 b	93 b	323 ab	18 b	9 b
Control	5000 g	2900 a	58 a	1977 d	90 a	33 a
F		34.05	50.92	1424.10	73.40	171.91
*p*<		0.001	0.001	0.001	0.001	0.001
LSD		430.0	7.032	62.73	12.36	2.761
SE		186.5	3.1	27.2	5.4	1.2
CV%		5.3	25.6	4.9	20.5	11.1

The values presented are the means of three replicates. LSD = least significant difference; CV% = coefficient of variance. Means followed by the same letter do not differ significantly at *p* ˂ 0.05 according to Duncan’s multiple range test.

## Data Availability

Not applicable.

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
