# Peer review of "Biocontrol of Maize Weevil, *Sitophilus zeamais* Motschulsky (Coleoptera: Curculionidae), in Maize over a Six-Month Storage Period"

_microorganisms, 2023, doi:10.3390/microorganisms11051261_

Round 1

Reviewer 1 Report (Previous Reviewer 3)

Beauveria bassiana strain MS-8 was used as a biocontrol agent to control maize weevil, Sitophilus zeamais Motschulsky (Coleoptera: Curculionidae), in maize over a six months storage period in this research. It is an interesting topic. The results showed that B. bassiana strain MS-8 induced a high level of mortality in S. zeamais, which subsequently reduced the levels of grain damage and grain weight loss. These results suggest that B. bassiana Strain MS-8 could be used as an alternative to chemical cotrol for the management of maize weevil infestations. However, there are still some questions that need to be clarified. 

1. I think a zero level B. bassiana (i.e., 100% kaolin) control should be run, even there some reports had shown that kaolin at 4 g kg−1 caused zero mortality levels in the target pests. Because the specific conditions of each experiment are different. Although the author explained the reason, I do not agree with it.

2. What's your definition of "edible grain"?

3. I suggest the author to provide the results of the identification of the fungus, the picture of colonies and conidia and deposit the ITS sequence in a public database (e.g. NCBI).

4. There is room for improvement in the experimental design of this study. The results showed that strain MS-8 applied in a kaolin level of 1 g/kg performed the best. Is 1 g/kg the best dosage? How about 0.5 1 g/kg? 

Some small corrections, please see the attachment.

Author Response

Kindly, find our responses in the attached file. 

Reviewer 2 Report (Previous Reviewer 2)

Dear authors,

you solved most of my previous comments. However, some minor inaccuracies remain, mainly in the presentation of average numbers in the result section. Please, see the points that I have highlighted in the pdf file.

In any case, after the suggested corrections, you paper is in my opinion worthy of publication.

Best regards

Author Response

Kindly, find our responses in the attached file. 

This manuscript is a resubmission of an earlier submission. The following is a list of the peer review reports and author responses from that submission.

Round 1

Reviewer 1 Report

There are too many grammatical errors to fix; thus, authors must hire an English editing service to improve the manuscript.

The title is too long and should be shortened.

Saeed 2017 is cited in the text however not available in the bibliography. I would recommend thoroughly checking all the citations and references.

Line 45: wrong citation. It should be Sinha & Sinha, 1991

Line 80. Truncated information

Line 110: 1 × 108 conidia ml−1???

Line 135: Why these specific doses were used? Did the authors conduct a pilot study first to determine the best concentration?

Line 152-153: what were the criteria for this longer duration (six months)? Fungi typically take about a week to cause mortality.

Line 195: Could you please explain the reason for a sudden increase in the percentage of live weevils (23%) in month five and then again decrease to 12%? Was there any other factor due to that insect surviving, or did the fungus lose its efficacy and then retain it? It seems a bit sloppy here.

I would like to suggest that the discussion should be developed by adding some logical points showing the reasons, not merely comparing one study with another, like in this manuscript.

The references cited should be from the last 2-5 years of published data. Most of them cited here were published ten or 20 years before.

Author Response

Kindly, receive my response regarding your comments 

Reviewer 2 Report

Dear authors,

the paper is well written and the work is clearly presented, and the topic is worthy of interest. However, there are some small inaccuracies that I have highlighted in the pdf file. Among these, you cannot state that all the mortality in the treatments is due to Beauveria bassiana, as other authors observed a synergistic effect of kaolin, such as

Storm, C.; Scoates, F.; Nunn, A.; Potin, O.; Dillon, A. Improving Efficacy of Beauveria bassiana against Stored Grain Beetles with a Synergistic Co-Formulant. Insects 2016, 7, 42. https://doi.org/10.3390/insects7030042. Please, consider this synergistic effect in your text.

In any case, after the suggested corrections, you paper is in my opinion worthy of publication.

Best regards

Author Response

(The authors gave the same response as above.)

Reviewer 3 Report

Beauveria bassiana strain MS-8 was used as a biocontrol agent to control maize weevil, Sitophilus zeamais Motschulsky (Coleoptera: Curculionidae), in maize over a six months storage period in this research. It is an interesting topic. The results showed that B. bassiana Strain MS-8 induced a high level of mortality in S. zeamais, which subsequently reduced the levels of grain damage and grain weight loss. These results suggest that B. bassiana Strain MS-8 could be used as an alternative to chemical cotrol for the management of maize weevil infestations. However, there are still some questions that need to be clarified. 

1. The title of the manuscript "Doubling the food supply in Africa" is too exaggerated. I suggest revising it. 

2. The manuscript does not mention whether Beauveria bassiana itself is infectious to maize. The Beauveria bassiana strain MS-8 used in this study is cultivated from rice grains, so whether it is infectious to maize is a very noteworthy issue.

3. Did you find any case that  Beauveria bassiana strain MS-8 grow out of the insect (Sitophilus zeamais) cadaver?

4. I think a zero level B. bassiana (i.e., 100% kaolin) control should be run, even there some reports had shown that kaolin at 4 g kg−1 caused zero mortality levels in the target pests. Because the specific conditions of each experiment are different.

5. Fig 1 dose not label well.

6. Fig 6 should be edited better. 

Author Response

(The authors gave the same response as above.)

Round 2

Reviewer 1 Report

My decision is to accept this manuscript with a minor change in the title.
Doubling the food supply in Africa, should be removed from the title as it seems exaggerated statement.

Author Response

Kindly, Find my response regarding your comments 

Reviewer 3 Report

The authors have made corresponding revisions to this manucript. Most questions were well revised or explained. However, I can not agree to the authors explain why they did not run a zero level control. And there are still some minor errors such as: "Salerno 151 et al. [32]". Please 

Author Response

Kindly, find my response regarding your comments 
